# The Effect of the Human Peptide GHK on Gene Expression Relevant to Nervous System Function and Cognitive Decline

**DOI:** 10.3390/brainsci7020020

**Published:** 2017-02-15

**Authors:** Loren Pickart, Jessica Michelle Vasquez-Soltero, Anna Margolina

**Affiliations:** Research & Development Department, Skin Biology, 4122 Factoria Boulevard SE Suite No. 200 Bellevue, WA 98006, USA; jessica@skinbiology.com (J.M.V.-S.); anna@amargolina.com (A.M.)

**Keywords:** GHK, copper, dementia, Alzheimer’s disease, Parkinson’s disease, neurons, glial cells, DNA repair, anti-oxidant, anti-anxiety, anti-pain

## Abstract

Neurodegeneration, the progressive death of neurons, loss of brain function, and cognitive decline is an increasing problem for senior populations. Its causes are poorly understood and therapies are largely ineffective. Neurons, with high energy and oxygen requirements, are especially vulnerable to detrimental factors, including age-related dysregulation of biochemical pathways caused by altered expression of multiple genes. GHK (glycyl-l-histidyl-l-lysine) is a human copper-binding peptide with biological actions that appear to counter aging-associated diseases and conditions. GHK, which declines with age, has health promoting effects on many tissues such as chondrocytes, liver cells and human fibroblasts, improves wound healing and tissue regeneration (skin, hair follicles, stomach and intestinal linings, boney tissue), increases collagen, decorin, angiogenesis, and nerve outgrowth, possesses anti-oxidant, anti-inflammatory, anti-pain and anti-anxiety effects, increases cellular stemness and the secretion of trophic factors by mesenchymal stem cells. Studies using the Broad Institute Connectivity Map show that GHK peptide modulates expression of multiple genes, resetting pathological gene expression patterns back to health. GHK has been recommended as a treatment for metastatic cancer, Chronic Obstructive Lung Disease, inflammation, acute lung injury, activating stem cells, pain, and anxiety. Here, we present GHK’s effects on gene expression relevant to the nervous system health and function.

## 1. Introduction

Age-related cognitive decline is a common problem for many elderly people, yet its cause is poorly understood. Over 99% of investigational drugs, participating in over 200 clinical trials, failed to receive approval for the treatment of Alzheimer’s disease [1]. Even the success of a few approved drugs provides only minimal patient improvement. There is a need for new, safe, and effective therapeutics with extensive safety and efficacy data that can be developed for use in humans within the next few years.

GHK (glycyl-l-histidyl-l-lysine) is a human plasma copper-binding peptide with a stunning array of actions that appear to counter aging-associated diseases and conditions. GHK was isolated in 1973 as an activity bound to human albumin that caused aged human liver tissue to synthesize proteins like younger tissue [2]. It has a strong affinity for copper and readily forms the complex GHK-Cu. It was first proposed that GHK-Cu functions by modulating copper intake into cells [3]. Since then, it has been established that the GHK peptide has stimulating and growth-promoting effects on many cells and tissues such as chondrocytes [4], liver cells and human fibroblasts [5]. It increases stemness and stimulates integrin secretion in human epidermal basal keratinocytes [6], as well as has a strong wound-healing and tissue-repairing effect [7]. GHK has also been shown to improve wound healing in controlled experiments using animals, such as rats, dogs, and rabbits [8,9,10].

In 2010, Hong et al. using the Broad Institute’s Connectivity Map (cMap), a compendium of transcriptional responses to compounds, identified GHK as the most active of 1309 bioactive substances, uniquely capable of reversing the expression of 54 genes in a metastatic-prone signature for aggressive early stage mismatch-repair colorectal cancer. GHK was active at a very low concentration of 1 µM [11].

Another study, which also used the cMap to identify genes affected by GHK, was conducted in 2012 when Campbell et al. identified 127 genes whose expression levels were associated with regional severity of chronic obstructive pulmonary disease (COPD). Emphysema and chronic bronchitis, the two main conditions of COPD, cause both small airway obstruction and significant loss of lung function over time. The cMap predicted that GHK would reverse the aberrant gene-expression signature associated with emphysematous destruction and induce expression patterns consistent with healing and repair. These finding were supported by laboratory experiments. GHK, at 10 nM, added to cultured fibroblasts from the affected lung areas of patients, changed gene expression patterns from tissue destruction to tissue repair. This led to the organization of the actin cytoskeleton, elevated the expression of integrin beta 1, and restored collagen contraction [12].

In addition to topping the list of 1309 biologically active molecules as the computer-recommended treatment for both human COPD (chronic obstructive pulmonary disease) and aggressive metastatic colon cancer, GHK has been recommended as a treatment for inflammations, acute lung injury, activation of stem cells, regeneration of aged skin, wound healing and tissue regeneration (skin, hair follicles, stomach and intestinal linings, hair growth, and boney tissue). It is also widely used in anti-aging skin products [13].

Even though it is not always possible to link gene expression data to biological actions, it is important to notice that GHK is highest in very healthy young people. Unfortunately, GHK declines with age. In studies at the University of California at San Francisco, young (age 20–25), male medical students were found to have about 200 nanograms/mL of GHK in their blood plasma, while the healthy, male medical school faculty (average age of 60) had only 80 nanograms/mL [7].

Our previous publication reviewed the biological effects of GHK relevant to neurodegeneration and cognitive health [14]. This paper will discuss the effect of GHK on gene expression relevant to nervous system functions and cognitive decline as well as review genetic and laboratory data relevant to nerve outgrowth, copper transport into cells, anxiety and pain, DNA repair, the ubiquitin proteasome system, the anti-oxidant system, changes in gene expression for glial cells, astrocytes, brain cells, dendrites, ganglia, motor neurons, Schwann cells, and sensory cells. It will also present possible methods for the use of therapeutic GHK in the treatment of nerve diseases.

## 2. Materials and Methods

The cMap was used to acquire the gene expression data. It is a large database that contains more than 7000 gene expression profiles of 5 human cell lines treated with 1309 distinct small molecules. Three GHK profiles are contained in this repository. The profiles are the result of cell lines treated with GHK at 1 micromolar which is around the concentration where many of GHK’s cellular effects occur [15]. These profiles were created using the GeneChip HT Human Genome U133A Array. Among the 5 cell lines used by the Connectivity Map only 2 were treated with GHK. Two of the profiles were created using the PC3 cell line - human prostate cancer cells, while the third used the MCF7 cell line – human breast cancer cells. Our studies utilized all three gene expression profiles.

GenePattern, a publicly available computational biology open-source software package developed for the analysis of genomic data, was used to analyze the gene data obtained from the cMap. The CEL files (data files used by Affymetrix software, used by the Broad Institute) were processed with MAS5 (Microarray Analysis Suite 5 software, Affymetrix, Santa Clara, CA, USA) and background correction. Files were then uploaded to the ComparativeMarkerSelectionViewer module in order to view fold changes for each probe set. Gene abbreviations appearing throughout the paper are consistent with the NCBI Gene database [16]. 

Due to multiple probe sets mapping to the same gene, the fold changes in m-RNA production produced by GenePattern were converted to percentages, and then all probe sets representing the same gene were averaged. It was determined that the 22,277 probe sets in the Broad data represent 13,424 genes. This ratio (1.66) was used to calculate the overall number of genes that affect GHK at various cutoff points (rather than probe sets).

The percentage of genes stimulated or suppressed by GHK with a change greater than or equal to 50% was estimated to be 31.2% [17]. Listed in the article are the gene expression effects of GHK on over 700 human genes associated with various nerve cell types. For well-defined systems where animal and cell cultures exist, such as anti-pain and anti-oxidation, relevant genes were manually chosen. For other systems, each gene’s Gene Ontology description was searched, using terms such as “neuron” or “glial”. The Gene Ontology consortium provides controlled vocabularies for the description of the molecular function, biological process, and cellular component of gene products. [18]. For most systems, gene expression numbers were given from 100% + or − and larger.

The cMap data was proven to be predictive of biological actions in most cases. In 2010, cMap predicted the anti-cancer actions of GHK. Subsequent work found GHK at 1 to 10 nanomolar reset the programmed cell death system on human nerve cancer cells and inhibited their growth in culture, while having the same effect on sarcoma cell growth in mice; it changed the gene expression of over 80 genes in an anti-growth manner [17]. Data from cMap also led to experiments that found GHK at 10 nanomolar caused human COPD-afflicted lung cells to switch cell expression from tissue destruction to repair and remodeling. For anti-oxidant actions, cMap has been very predictive of actions in mammals. However, gene expression numbers can vary widely at times and are not always predictable. For example, the cMap gives NGF (nerve growth factor) as a −243% decrease, yet in vivo rat studies have found NGF to be increased and two in vitro cell culture studies have found GHK to increase nerve outgrowth, an effect usually attributed to NGF.

Below, we cover GHK’s relationship with the following.
Nerve OutgrowthCopper Lack in Nerve DiseasesAnti-Anxiety and Anti-PainAnti-Oxidant Biological and Gene Expression DataDNA Repair Data and Gene Expression DNA RepairRestoring Regeneration after Cortisone TreatmentGene Expression—Clearing Damaged Protein with the Ubiquitin Proteasome System (UPS)Gene Expression—NeuronsGene Expression—Motor neuronsGene Expression—Glial cellsGene Expression—AstrocytesGene Expression—SchwannGene Expression—MyelinGene Expression—DendriteGene Expression—Oligodendrocyte cellsGene Expression—Schwann cellsGene Expression—SpinalPossible methods of therapeutic use of GHK for nerve disease

## 3. Results

### 3.1. Nerve Outgrowth

The lack of nerve outgrowth growth is considered a major factor in dementia [19,20,21].

GHK was discovered in 1973 as a growth factor for cultured hepatocytes. In 1975, Sensenbrenner and colleagues reported that GHK induced the formation chick embryonic neurons while suppressing glial cells. See Figure 1 [22].

Lindner and colleagues found that explants from chick embryo PNS (ganglion trigeminale) and from CNS of embryonal rats (hippocampus) and dissociated cells from chick embryo cerebral hemispheres that 0.01 microgram GHK per ml of medium stimulated the outgrowth of neuronal processes. Again, GHK promoted neuronal growth but not glial cells [23].

In studies of rats, severed sciatic nerves (axotomy) were inserted into a collagen prosthesis to which GHK was bonded. These were re-inserted into the rat, then removed after 10 days. GHK enhanced the production of trophic factors (Nerve Growth Factor, Neurotrophins 3 and 4) and recruited hematogenous cells and Schwann cells, which in turn help in the secretion of certain vital trophic and tropic factors essential for early regeneration. This improved nerve regeneration following axotomy [24]. Surprisingly, GHK’s gene expression data gives suppression of NGF (−243%) and NGFR (nerve growth factor receptor) (−132%). Thus, the biological system within wounded rat’s nervous tissue is more complex and probably due to other nerve stimulatory molecules.

### 3.2. Copper Deficiency, Dementia, and Nerve Dysfunction

Copper is an essential component of important anti-oxidant proteins such as SOD (copper zinc superoxide dismutase), ceruloplasmin, and Atox1 (human antioxidant protein 1). Copper deficiency has been postulated as a causative factor in a variety of gene diseases such as Alzheimer’s [25,26,27,28,29,30], myelopathy [31], motor neuron diseases and amyotrophic lateral sclerosis [32], Huntington’s [33], Lewy bodies and Creutzfeldt Jakob diseases [34].

More importantly, analysis of actual human brains from deceased patients with dementia has found the damaged areas to have very little cellular copper. In plaques from persons with Alzheimer’s disease, iron and aluminum appear to cause plaque formation while copper and zinc may be protective [26,27,28,35,36,37].

Copper deficiency caused by bariatric surgery or gastrointestinal bleeding led to myelopathy (human swayback), paralysis, blindness and behavioral and cognitive changes. Mice born and maintained on a copper deficient diet had 80% reduction in brain copper level at 6-8 weeks and had neuronal and glial changes typical for neurodegenerative disorders [25,31,38,39].

#### 3.2.1. Supplying Copper to Nerve Cells

Though copper deficiency appears linked to major nerve diseases, the use of copper supplements as a treatment has been disappointing. A placebo-controlled study of 68 Alzheimer’s patients (34 control, 34 copper) with a treatment of 8 mgs of daily copper (a high level) for 1 year produced no negative findings. This seems to rule out excessive copper levels as a causative agent for the development of Alzheimer’s. The predictive protein marker, CSF Abeta42, is lower in persons developing Alzheimer’s. Subjects given extra copper supplementation maintained this protein at a higher level, a possible positive effect, but there was minimal improvement in the disease [40].

One small copper complex chelator, CuATSM (diacetyl-bis(4-methylthiosemicarbazonato)copper 2+), has given indications of ameliorating the effects of ALS (familial amyotrophic lateral sclerosis) in a strain of genetically modified mice that develop a form of ALS. CuATSM extends life in such mice by up to 25%. The motor neuron disease can be restarted and then stopped by controlling CuATSM treatment. The treatment increases the amount of active superoxide dismutase in the mice [41]. The safety of CuATSM is largely unknown. The safety data sheet states the following: “Material may be irritating to the mucous membranes and upper respiratory tract. May be harmful by inhalation, ingestion, or skin absorption. May cause eye, skin, or respiratory system irritation. To the best of our knowledge, the toxicological properties have not been thoroughly investigated.”

GHK-Copper 2+ increased superoxide dismutase (SOD) activity in mice as detailed below in Section 4 [42].

#### 3.2.2. Albumin, GHK and Copper Transport

Both albumin and GHK transport copper 2+ to cells and tissues. However, in human blood, there are 700 albumin molecules for each GHK molecule, so albumin is the major source of copper for tissue use. GHK and albumin have high and very similar binding constants for copper 2+ (Albumin = pK binding log 10 |16.2|; GHK = pK binding log 10 |16.4|). Human plasma contains about 15 micromolar copper and 12% (1.8 micromolar) of this is bound to albumin. But GHK-Cu is maximally active on most cells around one nanomolar or less. Aqueous dialysis studies established that GHK can obtain copper 2+ from albumin. We assume that this also occurs in cell culture and within mammals and that GHK has adequate copper for biological actions.

Our studies over the past 39 years have indicated that virtually all biological GHK effects require the presence of copper 2+ chelated to the tripeptide. Strong copper chelators such as bathocuproine abolish GHK actions. GHK alone is often effective in murine wound healing or hair growth models, but GHK-Cu always produced much stronger responses. GHK attached to radioactive copper-64 increases copper uptake into cultured hepatoma cells [7].

The intravenous injection of tritiated copper-free GHK into mice was found, after 4 h, to concentrate most densely within the animals’ kidneys and brain. See Figure 2 [43].

The best evidence that GHK can obtain copper 2+ from body fluids was from a study that used biotinylated GHK bound to collagen films placed over wounds in rats. The GHK pads raised the copper concentration by ninefold at the wound site when compared to non-GHK collagen films. Such biotinylated GHK collagen films also increased wound healing, cell proliferation, and increased the expression of antioxidant enzymes in the treated group [9].

Most importantly, GHK activates numerous regenerative and protective genes. Albumin will not mimic the GHK activated systems. So GHK must act through a separate pathway, not the albumin pathway. Albumin’s copper feeds cells; GHK’s copper activates regenerative and protective genes.

GHK-Cu’s regenerative and protective actions on tissue are very similar to those found by John R Sorenson throughout his 33 years of work on various copper salicylates. See Table 1. It appears that GHK-Copper and Sorenson’s DIPS-Cu (diisopropylsalicylate-copper 2+) both activate the same pathway, a pathway strongly associated with tissue health and repair. GHK-copper 2+ (molecular weight 404) and Sorenson’s DIPS-Cu (molecular weight 506) are both very small molecules while albumin is much larger (molecular weight 64,000). Hence, they are likely to use different cell receptor systems [44,45,46,47,48,49]. See Figure 3.

### 3.3. Anti-Anxiety (Anxiolytic) and Anti-Pain

Anxiety and pain are serious issues in patients with dementia and other disabling mental conditions. Opiate peptides often possess both anti-pain and wound healing properties [50]. When healthy human males were fed a low copper diet (1 mg/day of copper) for 11 weeks, their plasma opiate levels dropped by 80%. As soon as copper was restored (with a diet containing 3 mg/day of copper), the levels returned to normal [51].

GHK has been found to possess analgesic and anxiolytic effects (anti-anxiety) in animal experiments. GHK reduced pain after thermal injury to rats at a dose of 0.5 milligrams/kg. Within 12 min after intraperitoneal injection, it also increased the amount of time the rats spent exploring more open areas of the maze and decreased time spent immobile (the freeze reaction), which indicated reduction of fear and anxiety. These effects were observed at 0.5 micrograms/kg [52,53].

These effects also prove that GHK rapidly affects the brain perception and function. This is an area where GHK could be used on patients today.

A manual search of genes affected by GHK found that seven anti-pain genes increased and two genes decreased. See Table 2 and Table 3.

### 3.4. Antioxidant Activity of the GHK Peptide

High metabolic activity found in the brains of both humans and animals results in elevated oxygen consumption and constant production of reactive oxygen species (ROS) in mitochondria. At the same time, the brain tissue is rich in unsaturated fatty acids and transition metal ions, yet has relatively fewer antioxidants compared to other organs, creating favorable conditions for oxidative damage. Since the blood-brain barrier prevents many dietary antioxidants from entering the brain, it largely relays on endogenous antioxidants such as Cu and Zn dependent superoxide dismutase (Cu, Zn SOD1). This enzyme requires the metal ions copper and zinc in order to be active. Hence, copper deficiency can lead to reduced SOD activity and increased oxidative brain damage. When pregnant rats were fed a copper deficient diet, the embryos displayed low SOD activity, increased super oxide anion radical level, and higher incidence of DNA damage and malformations [54].

GHK has broad and powerful anti-oxidation properties in both mammals and cell culture, and it is known to increase anti-oxidant gene expression. Tissue oxidation has been postulated as a causative factor in Parkinson’s disease and other various nerve diseases of aging [55,56,57,58,59].

Diminished copper has been found in cells expressing SOD1 mutations postulated to cause ALS in mice and increase memory loss [60,61].

A peptidomimetic inhibitor (P6), based on GHK, interacts with amyloid beta (Aβ) peptide and its aggregates. P6 prevents the formation of toxic Aβ oligomeric species, fibrillar aggregates and DNA damage. It is a potential therapeutic candidate to ameliorate the multifaceted Aβ toxicity in Alzheimer’s [62].

#### 3.4.1. GHK’s Anti-Oxidant Effects in Mammals and Cell Culture

The use of GHK-Cu in mice protected their lung tissue from lipopolysaccharide-induced acute lung injury (ALI). When GHK-Cu was used by mice with LPS-induced ALI, it attenuated related histological alterations in the lungs and suppressed the infiltration of inflammatory cells into the lung parenchyma. The GHK-Cu also increased superoxide dismutase (SOD) activity while decreasing TNF-α and IL-6 production through the suppression of the phosphorylation of NF-κB p65 and p38 MAPK in the nucleus of lung cells [42].

P38 mitogen-activated protein kinases are responsive to stress stimuli, such as cytokines, ultraviolet irradiation, heat shock, and osmotic shock, and are involved in cell differentiation, apoptosis, and autophagy. NF-κB/RELA p65 activation has been found to be correlated with cancer development, suggesting the potential of RELA as a cancer biomarker. Specific modification patterns of RELA have also been observed in many cancer types.

Multiple antioxidant actions of GHK have been demonstrated in vitro and in animal wound healing studies. They include inhibiting the formation of reactive carbonyl species (RCS), detoxifying toxic products of lipid peroxidation such as acrolein, protecting keratinocytes from lethal UVB radiation, and preventing hepatic damage by dichloromethane radicals.

The ability of GHK to prevent oxidative stress was tested in vitro using Cu(2+)-dependent oxidation of low-density lipoproteins (LDL). LDL were treated with 5 μM Cu(2+) for 18 h in either phosphate buffered saline (PBS) or Ham’s F-10 medium. There was increased production of thiobarbituric acid reactive substances (TBARSs), which indicated increased oxidation. GHK and histidine “entirely blocked” (quoted from the article) the in vitro Cu(2+)-dependent oxidation of low-density lipoproteins (LDL). In comparison, superoxide dismutase (SOD1) provided only 20% reduction of oxidation [63].

Acrolein, a well-known carbonyl toxin, is produced by lipid peroxidation of polyunsaturated fatty acids. GHK effectively reduces the formation of both acrolein and another product of oxidation, 4-hydroxynonenal. GHK also blocks lethal ultraviolet radiation damage to cultured skin keratinocytes by binding and inactivating reactive carbonyl species such as 4-hydroxynoneal, acrolein, malondialdehyde, and glyoxal [64,65,66].

The intraperitoneal injection of 1.5 mg/kg of GHK into rats for five days before dichloromethane poisoning and five days thereafter provided protection of the functional activity of hepatocytes and immunological responsiveness. Dichloromethane is toxic to hepatic tissue via the formation of a dichloromethane free radical that induces acute toxic damage [67].

In rats with experimental bone fractures peptides, GHK (0.5 μg/kg), dalargin (1.2 μg/kg), and thymogen (0.5 μg/kg) were injected intraperitoneally. Within 10 days, there was a decrease of malonic dialdehyde and an increase of catalase activity in blood. There was also a marked increase of reparative activity. Each combination of peptides was more potent than any of the studied peptides injected separately. The synergetic action of peptides Gly-His-Lys, thymogen, and dalargin was proposed for stimulation of reparative osteogenesis [68].

GHK-Cu reduced iron release from ferritin by 87%. Iron has also been shown to have a direct role in the initiation of lipid peroxidation. An Fe(2+)/Fe(3+) complex can serve as an initiator of lipid oxidation. In addition, many iron complexes can catalyze the decomposition of lipid hydroperoxides to the corresponding lipid alkoxy radicals. The major storage site for iron in serum and tissue is ferritin. Ferritin in blood plasma can store up to 4500 atoms of iron per protein molecule, and superoxide anions can promote the mobilization of iron from ferritin. This free iron may then catalyze lipid peroxidation and the conversion of a superoxide anion to the more damaging hydroxyl radical [69].

#### 3.4.2. Synthesis of GHK-Cu Analogs with Higher Anti-ROS Activity

GHK-Cu has, on a molar basis, about 1% to 3% of the activity of the Cu, Zn superoxide dismutase protein. By simple modifications to the peptide, it is possible to raise the SOD-mimetic activity up 223-fold. Given the broad range of the antioxidant actions of GHK, it is likely that modifications will increase its countering reactive species such as RCS and dichloromethane radicals. See Table 4 [70].

#### 3.4.3. Antioxidant Gene Expression Analysis

A manual search of antioxidant associated genes effected by GHK yielded 18 genes with significant antioxidant activity. See Table 5 and Table 6.

### 3.5. DNA Repair, Cell Culture, and Gene Expression

A lack of adequate DNA repair may be related to neurological degeneration in the aging population [90,91,92,93].

DNA damage is a major problem in the life cycle of biological cells. Normal cellular metabolism releases compounds that damage DNA such as reactive oxygen species, reactive nitrogen species, reactive carbonyl species, lipid peroxidation products and alkylating agents, among others, while hydrolysis cleaves chemical bonds in DNA. It is estimated that each normally functioning cell in the human body suffers at least 10,000 DNA damaging incidents daily [94].

Radiation therapy is believed to stop cell replication by damaging cellular DNA. A study of cultured primary human dermal fibroblast cell lines from patients who had undergone radiation therapy for head and neck cancer found that the procedure slowed the population doubling times for the cells. But treatment with one nanomolar GHK-Cu restored population doubling times to normal. Irradiated cells treated with GHK-Cu also produced significantly more basic fibroblast growth factor and vascular endothelial growth factor than untreated irradiated cells [5].

GHK is primarily stimulatory for gene expression of DNA Repair genes (47 UP, 5 DOWN), suggesting an increased DNA repair activity. Here we searched the Gene Ontology descriptions for “DNA Repair”. See Table 7 and Table 8.

### 3.6. Restoring Regeneration After Cortisone Treatment

Steroid dementia syndrome describes the signs and symptoms of hippocampal and prefrontal cortical dysfunction, such as deficits in memory, attention, and executive function, induced by glucocorticoids. Dementia-like symptoms have been found in some individuals who have been exposed to glucocorticoid medication, often dispensed in the form of asthma, arthritis, and anti-inflammatory steroid medications. The condition reverses, but not always completely, within months after steroid treatment is stopped [95].

In the human body, cortisone and cortisol are easily interconvertible and have similar anti-inflammatory actions. They also profoundly inhibit tissue regeneration, such as wound repair. DHEA (dehydroepiandrosterone) is an androgenic hormone. It is a precursor for testosterone and the estrogens. DHEA antagonizes the effects of cortisol but decreases about 80% from age 20 to age 80 while cortisone/cortisol levels remain high. It has been proposed that many of the deleterious effects of aging are due to excessive cortisol that is not balanced by DHEA.

GHK-Cu, when administered systemically to mice, rats, and pigs, counters the wound healing inhibition of cortisone throughout the animal [96].

### 3.7. Gene Expression—Clearing Damaged Protein—Ubiquitin Proteasome System

The ubiquitin proteasome system (UPS) clears damaged proteins. Insufficient activity of this system is postulated to produce an accumulation of toxic protein oligomers which start the neurodegenerative process. During aging, there is decreased activity of the ubiquitin proteasome system. To date, no effective therapies have been developed that can specifically increase the UPS activity [97,98,99,100].

GHK strongly stimulates the gene expression of the UPS system with 41 genes increased and 1 gene suppressed. Here we searched gene title for “ubiquitin” or “proteasome”. See Table 9 and Table 10.

### 3.8. Gene Expression—Neurons

Neurons are cells that carry messages between the brain and other parts of the body; they are the basic units of the nervous system.

GHK is primarily stimulatory for gene expression of neuron related genes. Here we searched the Gene Ontology descriptions for “Neuron”. See Table 11 and Table 12.

### 3.9. Motor Neurons

Motor neurons are nerve cells forming part of a pathway along which impulses pass from the brain or spinal cord to a muscle or gland.

Here we searched Gene Ontology descriptions for “motor neuron”. See Table 13 and Table 14.

### 3.10. Gene Expression—Glial Cells

Glial cells are non-neuronal cells that maintain homeostasis, form myelin, and provide support and protection for neurons in the central and peripheral nervous systems. 

Here we searched Gene Ontology descriptions for “glial”. See Table 15 and Table 16.

### 3.11. Astrocyte

Astrocytes are characteristic star-shaped glial cells in the brain and spinal cord. The astrocyte proportion varies by region and ranges from 20% to 40% of all glial cells. They perform many functions, including biochemical support of endothelial cells that form the blood–brain barrier, provision of nutrients to the nervous tissue, maintenance of extracellular ion balance, and a role in the repair and scarring process of the brain and spinal cord following traumatic injuries.

Here we searched Gene Ontology descriptions for “astrocyte”. See Table 17 and Table 18.

### 3.12. Schwann Cells

Schwann cells are cells of the peripheral nervous system that wrap around a nerve fiber, jelly-roll fashion, forming the myelin sheath.

Here we searched Gene Ontology descriptions for “Schwann”. See Table 19 and Table 20.

### 3.13. Myelin

Myelin is a mixture of proteins and phospholipids that form a whitish insulating sheath around many nerve fibers, increasing the speed at which impulses are conducted.

Here we searched Gene Ontology descriptions for “myelin”. See Table 21 and Table 22.

### 3.14. Gene Expression—Dendrites

Dendrites are short branched extensions of a nerve cell, along which impulses received from other cells at synapses are transmitted to the cell body.

Here we searched Gene Ontology descriptions for “dendrite”. See Table 23 and Table 24.

### 3.15. Gene Expression—Oligodendrocytes

Oligodendrocytes are glial cells similar to astrocytes, but with fewer protuberances, which are concerned with the production of myelin in the central nervous system.

Here we searched Gene Ontology descriptions for “oligodendrocyte”. See Table 25 and Table 26.

### 3.16. Gene Expression—Sensory Nerve cells

Sensory neurons are nerves that transmit sensory information (sight, sound, feeling, etc.). They are activated by sensory input and send projections to other elements of the nervous system, ultimately conveying sensory information to the brain or spinal cord.

Here we searched Gene Ontology descriptions for “sensory”. See Table 27 and Table 28.

### 3.17. Spinal Nerve Cells

Spinal nerve cells transfer information, which travels down the spinal cord, as a conduit for sensory information in the reverse direction, and finally as a center for coordinating certain reflexes.

Here we searched Gene Ontology descriptions for “spinal”. See Table 29 and Table 30.

## 4. Possible Methods of Therapeutic Use of GHK for Nerve Diseases

### 4.1. Mode of Administering GHK-Cu to Patients

#### 4.1.1. Skin Cream or Patch

GHK-Cu has an unexpectedly rapid passage through skin’s stratum corneum. When tested by Howard Maibach’s group (Univerisity of California at San Francisco), 0.68% GHK-Cu was applied to dermatomed skin. Over 48 h, 136 micrograms of GHK-Cu passed through the skin per centimeter squared. This is a significant amount of GHK-Cu, and a transdermal patch of a several centimeters squared may pass therapeutically effective amounts throughout the human body [101].

Russian studies reported that 0.5 micrograms/kg reduced anxiety in rats. Scaled up for a human weight of 70 kg, this would be 35 micrograms in a human [52]. Our studies on activation of systemic healing in mice, rats, and pigs suggest that about 50 milligrams of GHK-Cu would be effective throughout the human body, although dose-ranging to determine the minimum active dosage was never performed.

#### 4.1.2. Liposomal Encapsulated Oral Tablet

Alternately, the use of encapsulated liposomal GHK-Cu would allow its oral administration at relatively high dosages. Some sellers of an encapsulated liposomal tripeptide glutathione claim that 60% of the orally administrated peptide enters the human blood stream [102]. Direct administration in a regular pill form is unlikely to work because of GHK’s extreme sensitivity to breakdown by intestinal carboxypeptidase [103].

GHK-Cu costs about $8/gram in kilogram amounts. For a 50 mg dosage, the GHK-Cu would cost about $0.40. It is possible that GHK alone would be effective in humans and be able to obtain sufficient amounts of copper 2+ from albumin. If so, this would simplify its therapeutic use. The minimum effective dosage of GHK-Cu for various uses is unknown since such studies were never performed.

GHK-Cu does lower blood pressure, but the LD50 (Lethal Dose for 50% of mice) for such effects would be about a single dosage of 23,000 mgs of GHK-Cu in a 70 kg human. In GHK-Cu’s long history of use in cosmetics, no health issues have ever arisen. We were never able to find an LD 50 for GHK without copper.

In our studies, equimolar mixtures of GHK-Cu and GHK (no copper) are often used to avoid any release of loosely bound copper. Also, copper chelators such as penicillamine have been reported to cause psychosis in humans [104].

## 5. Conclusions

Given all the failed attempts to develop effective treatment methods for nerve degeneration, it is suggested that researchers must take a very broad view of the possible factors causing neurodegenerative diseases and not focus on limited possible causes. It is sensible to concentrate research efforts on the reversion of affected tissues to a healthier condition more characteristic of younger humans. GHK gene studies have increasingly led to the conclusion that the conditions and diseases of aging cannot be scientifically treated without understanding the extensive changes in overall gene activity during aging. 

There are three sources of evidence on GHK actions:
The best data is in vivo mammalian data, including human clinical studies. As reviewed in this paper, these studies give overwhelming evidence of GHK’s effects on cells and tissue growth, as well as anti-cancer, anti-oxidant, wound-healing, anti-inflammation, anti-pain, anti-anxiety and skin regeneration actions.A second form of data is in vitro cell culture and organ culture results. Culture results give evidence about the effect of GHK on cellular production of collagen and other structural proteins, the effect on stem cell function, the recovery of cellular function after anticancer radiation or ultraviolet radiation, and sensitivity of cells to oxidative molecules.A third source of data is in Human Gene expression. Data analysis found that GHK induces a 50% or greater (plus or minus) change of expression in 31.2% of human genes, affecting genes linked to multiple biochemical pathways in many organs and tissue, including the nervous system.

Many studies highlight gene expression effects of various molecules. Given today’s advances in computer modeling, it is not that difficult to find substances which affect gene expression in one way or another. However, in most cases, computer-based predictions do not have the same supporting evidence of in vivo and in vitro laboratory data as GHK has. Also, in many cases, the safety and cost of the proposed treatments are a big concern. GHK is safe, inexpensive, and can be used in humans today. 

The future research should be focused on further making sense of the very extensive gene data, which has to be paralleled with laboratory and clinical studies. GHK has a wealth of biological data in the areas of wound healing, hair and skin regeneration, intestinal tract and bone repair. However, there is a surprising lack of GHK research in the area of neurodegeneration and cognitive health. We hope that our gene data will encourage researchers to take a better look at biological actions and significance of GHK in connection with cognitive health and nervous system function. 

The best administration method, in our opinion, would be GHK-Cu incorporated into liposomes, then administered as an enteric capsule for oral use. A dosage of 10 mgs per dose would be a good starting point, at least for safety studies, but inducing positive actions will most likely require a higher dosage.

## Figures and Tables

**Figure 1 brainsci-07-00020-f001:**
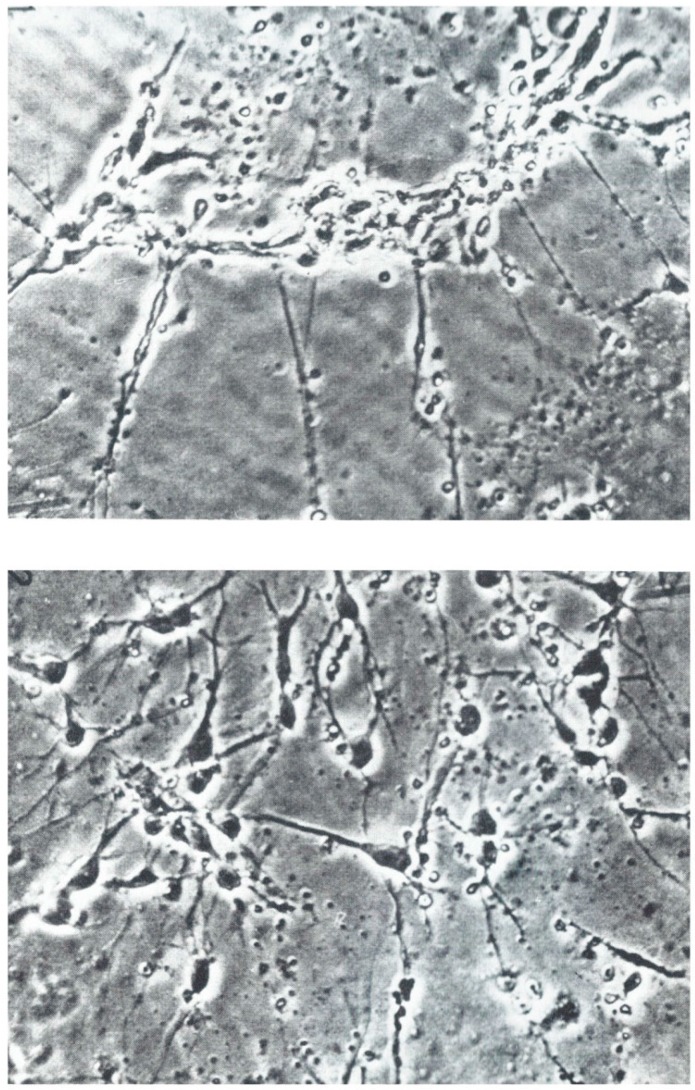
(**Top**)—Control; (**Bottom**)—Addition of 200 ng/mL of GHK to culture media (Phase contrast ×250, photo micrographs used with permission of John Wiley and Sons).

**Figure 2 brainsci-07-00020-f002:**
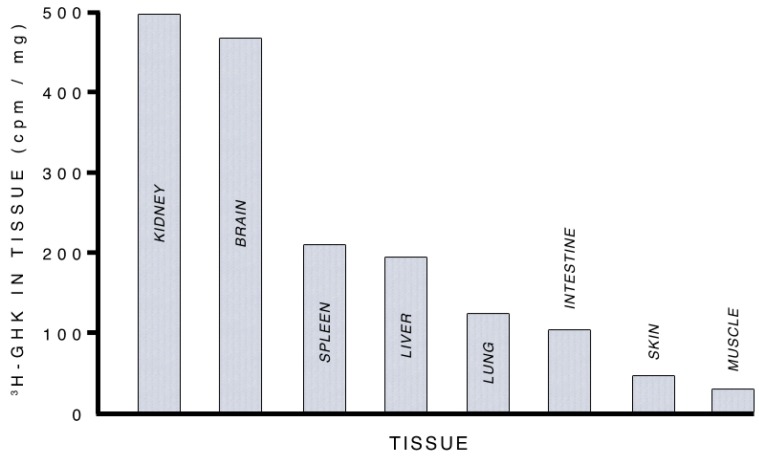
Uptake of glycyl-l-histidyl-l-lysine (GHK) into various mouse tissues. (Reprinted from Pickart, L. [43]).

**Figure 3 brainsci-07-00020-f003:**
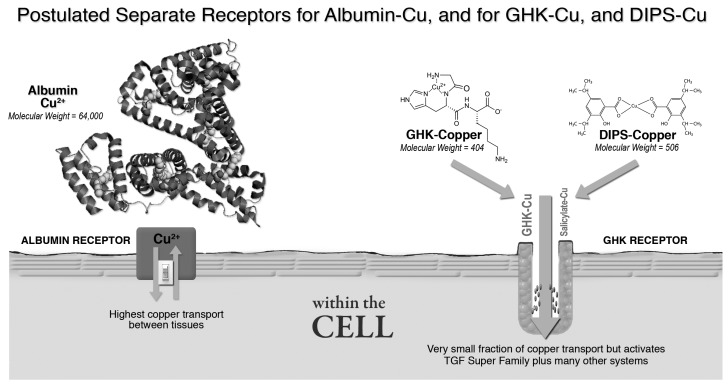
Proposed cell receptor for GHK-Cu.

**Table 1 brainsci-07-00020-t001:** Similarity of Actions of GHK-Copper and Diisopropylsalicylate-Copper.

Action	GHK-Copper 2+	Diisopropylsalicylate-Copper 2+
Wound Healing	Yes	Yes
Inhibit Cancer Growth	Yes	Yes
Anti-Ulcer	Yes	Yes
Anti-Pain	Yes	Yes
Improve Recovery After Radiation	Yes	Yes
Increase Stem Cell Activity	Yes	Yes

**Table 2 brainsci-07-00020-t002:** Distribution of Genes Affected by GHK and Associated with Pain.

Percent Change in Gene Expression	Genes UP	Genes DOWN
50%–99%	0	0
100%–199%	5	2
200%–299%	1	0
300%–399%	0	0
400%–499%	0	0
500%+	1	0
Total	7	2

**Table 3 brainsci-07-00020-t003:** GHK and Genes Associated with Pain.

**UP**	**Gene**	**Percent Change in Gene Expression**	**Comments**
1	OPRMI	1294	Opioid mu 1-High Affinity for enkephalins and beta-endorphins
2	OPRL1	246	Receptor for neuropeptide nociceptin
3	CCKAR	190	Cholecystokinin A receptor, cholecystokinin affects satiety, release of beta-endorphin and dopamine
4	CNR1	172	Cannabinoid receptor, pain-reducing
5	SIGMAR1	155	Non-opioid receptor
6	PNOC	150	Prepronociceptin, complex interactions with pain and anxiety induction
7	OXT	136	Ocytocin, bonding protein—gene also increases human chorionic gonadotropin
**DOWN**	**Gene**	**Percent Change in Gene Expression**	**Comments**
1	AMPA 3/GRIA3	−126.00%	Glutamate receptor, retrograde endocannaboid signaling, nervous system
2	OPRK1	−119.00%	Reduced cocaine effects

**Table 4 brainsci-07-00020-t004:** Superoxide Dismutase Mimetic Activity of GHK and Analogs.

Molecule	Superoxide Dismutase Mimetic Activity
Gly-His-Lys:Cu(2+)	100
Lys-His-Gly-Amide:Cu(2+)	21
Gly-His-Lys-Ala-Phe-Ala:Cu(2+)	561
Ala-His-Lys:Cu(2+)	563
Gly-His-Lys-Octyl Ester:Cu(2+)	810
Gly-His-Caprolactam:Cu(2+)	4500
His-Gly-Lys:Cu(2+)	22,300

**Table 5 brainsci-07-00020-t005:** Distribution of Genes Affected by GHK with Antioxidant Activity.

Percent Change in Gene Expression	Genes UP	Genes DOWN
50%–99%	2	0
100%–199%	7	1
200%–299%	2	0
300%–399%	1	0
400%–499%	1	0
500%+	3	1
Total	16	2

**Table 6 brainsci-07-00020-t006:** GHK and Genes Associate with Antioxidant Activity.

**UP**	**Genes**	**Percent Change in Gene Expression**	**Comments**
1	TLE1	762	Inhibits the oxidative/inflammatory gene NF-κB [71].
2	SPRR2C	721	This proline-rich, antioxidant protein protects outer skin cells from oxidative damage from reactive oxygen species (ROS). When the ROS level is low, the protein remains in the outer cell membrane, but when the ROS level is high, the protein clusters around the cell’s DNA to protect it [72,73].
3	ITGB4	609	Up-regulation of ITGB4 promotes wound repair ability and antioxidative ability [74].
4	APOM	403	Binds oxidized phospholipids and increases the antioxidant effect of high-density lipoproteins (HDL) [75].
5	PON3	319	Absence of PON3 (paraoxonase 3) in mice resulted in increased rates of early fetal and neonatal death. Knockdown of PON3 in human cells reduced cell proliferation and total antioxidant capacity [76].
6	IL18BP	295	The protein encoded by this gene is an inhibitor of the pro-inflammatory cytokine IL18. IL18BP abolished IL18 induction of interferon-gamma (IFN gamma), IL8, and activation of NF-κB in vitro. Blocks neutrophil oxidase activity [77].
7	HEPH	217	Inhibits the conversion of Fe(2+) to Fe(3+). HEPH increases iron efflux, lowers cellular iron levels, suppresses reactive oxygen species production, and restores mitochondrial transmembrane potential [78].
8	GPSM3	193	Acts as a direct negative regulator of NLRP3. NLRP3 triggers the maturation of the pro-inflammatory cytokines IL-1β and IL-18 [79].
9	FABP1	186	Reduces intracellular ROS level. Plays a significant role in reduction of oxidative stress [80,81].
10	AGTR2	171	AGTR2 exerts an anti-inflammatory response in macrophages via enhanced IL-10 production and ERK1/2 phosphorylation, which may have protective roles in hypertension and associated tissue injury [82].
11	PON1	149	PON1 (paraoxonase 1) is a potent antioxidant and a major anti-atherosclerotic component of HDL [83].
12	MT3	142	Metallothioneins (MTs) display in vitro free radical scavenging capacity, suggesting that they may specifically neutralize hydroxyl radicals. Metallothioneins and metallothionein-like proteins isolated from mouse brain act as neuroprotective agents by scavenging superoxide radicals [84,85].
13	PTGS2	120	Produces cyclooxygenase-II (COX-II), which has antioxidant activities [86].
14	SLC2A9	117	The p53-SLC2A9 pathway is a novel antioxidant mechanism. During oxidative stress, SLC2A9 undergoes p53-dependent induction, and functions as an antioxidant by suppressing ROS, DNA damage, and cell death [87].
**DOWN**	**Genes**	**Percent Change in Gene Expression**	**Comments**
1	IL17A	−1018	This cytokine can stimulate the expression of IL6 and cyclooxygenase-2 (PTGS2/COX-2), as well as enhance the production of nitric oxide (NO). High levels of this cytokine are associated with several chronic inflammatory diseases including rheumatoid arthritis, psoriasis, and multiple sclerosis ([88]).
2	TNF	−115	GHK suppresses this pro-oxidant TNF gene [89].

**Table 7 brainsci-07-00020-t007:** Distribution of Genes Affected by GHK and Associated with DNA Repair.

Percent Change in Gene Expression	Genes UP	Genes DOWN
50%–100%	41	4
100%–150%	2	1
150%–200%	1	0
200%–250%	2	0
250%–300%	1	0
Total	47	5

**Table 8 brainsci-07-00020-t008:** GHK and Genes Associate with DNA Repair.

**UP**	**Gene Title**	**Percent Change in Gene Expression**
1	poly (ADP-ribose) polymerase family, member 3, PARP3	253
2	polymerase (DNA directed), mu, POLM	225
3	MRE11 meiotic recombination 11 homolog A MRE11A	212
4	RAD50 homolog (S. cerevisiae), RAD50	175
5	eyes absent homolog 3 (Drosophila), EYA3	128
6	retinoic acid receptor, alpha, RARA	123
**DOWN**	**Gene Title**	**Percent Change in Gene Expression**
1	cholinergic receptor, nicotinic, alpha 4, CHRNA4	−105

**Table 9 brainsci-07-00020-t009:** Distribution of Genes Affected by GHK and Associated with the Ubiquitin Proteasome System.

Percent Change in Gene Expression	Genes UP	Genes DOWN
50%–99%	31	1
100%–199%	7	0
200%–299%	0	0
300%–399%	1	0
400%–499%	1	0
500%+	1	0
Total	41	1

**Table 10 brainsci-07-00020-t010:** GHK and Genes Associated with the Ubiquitin Proteasome System.

UP	Gene Title	Percent Change
1	ubiquitin specific peptidase 29, USP29	1056
2	ubiquitin protein ligase E3 component n-recognin 2, UBR2	455
3	gamma-aminobutyric acid (GABA) B receptor, 1 /// ubiquitin D, GABBR1 /// UBD	310
4	ubiquitin specific peptidase 34, USP34	195
5	parkinson protein 2, E3 ubiquitin protein ligase (parkin), PARK2	169
6	ubiquitin-conjugating enzyme E2I (UBC9 homolog, yeast), UBE2I	150
7	ubiquitin protein ligase E3 component n-recognin 4, UBR4	146
8	ubiquitin protein ligase E3B, UBE3B	116
9	ubiquitin specific peptidase 2, USP2	104
10	ubiquitin-like modifier activating enzyme 6, UBA6	104

**Table 11 brainsci-07-00020-t011:** Distribution of Genes Affected by GHK and Associated with Neurons.

Percent Change in Gene Expression	Genes UP	Genes DOWN
50%–99%	230	80
100%–199%	99	80
200%–299%	45	35
300%–399%	19	14
400%–499%	9	10
500%+	6	11
Total	408	230

**Table 12 brainsci-07-00020-t012:** GHK and Genes Associated with Neurons.

**UP**	**Gene Title**	**Percent Change**
1	opioid receptor, mu 1, OPRM1	1294
2	tumor protein p73, TP73	938
3	potassium voltage-gated channel, Shal-related subfamily, member 1, KCND1	845
4	solute carrier family 8 (sodium/calcium exchanger), member 2, SLC8A2	737
5	contactin associated protein-like 2, CNTNAP2	581
6	stathmin-like 3, STMN3	500
7	latrophilin 3, LPHN3	494
8	angiopoietin 1, ANGPT1	487
9	synapsin III, SYN3	478
10	dipeptidyl-peptidase 6, DPP6	448
11	somatostatin receptor 2, SSTR2	442
12	G protein-coupled receptor, family C, group 5, member B, GPRC5B	431
13	sodium channel, voltage-gated, type III, alpha subunit, SCN3A	423
14	smoothened homolog (Drosophila), SMO	415
15	tryptophan hydroxylase 1, TPH1	409
16	caspase 8, apoptosis-related cysteine peptidase, CASP8	399
17	gamma-aminobutyric acid (GABA) A receptor, alpha 5 /// gamma-aminobutyric acid receptor subunit alpha-5-like, GABRA5 /// LOC100509612	392
18	transcription factor 7 (T-cell specific, HMG-box), TCF7	372
19	solute carrier family 17 (sodium-dependent inorganic phosphate cotransporter), member 6, SLC17A6	369
20	doublecortin-like kinase 1, DCLK1	365
21	p21 protein (Cdc42/Rac)-activated kinase 1, PAK1	363
22	neurogenic differentiation 4, NEUROD4	362
23	zinc finger protein 335, ZNF335	358
24	wingless-type MMTV integration site family, member 3, WNT3	352
25	ADAM metallopeptidase domain 8, ADAM8	352
26	neuropeptide Y, NPY	346
27	potassium voltage-gated channel, Shaw-related subfamily, member 3, KCNC3	332
28	EPH receptor B1, EPHB1	330
29	LIM domain kinase 1, LIMK1	322
30	myeloid/lymphoid or mixed-lineage leukemia (trithorax homolog, Drosophila), MLL	318
31	growth associated protein 43, GAP43	305
32	FBJ murine osteosarcoma viral oncogene homolog, FOS	305
33	sal-like 1 (Drosophila), SALL1	302
34	synovial sarcoma, X breakpoint 2 /// synovial sarcoma, X breakpoint 2B, SSX2 /// SSX2B	301
35	inositol 1,4,5-triphosphate receptor, type 3, ITPR3	298
36	bone morphogenetic protein receptor, type IB, BMPR1B	298
37	synuclein, gamma (breast cancer-specific protein 1), SNCG	292
38	calcium channel, voltage-dependent, P/Q type, alpha 1A subunit, CACNA1A	286
39	capping protein (actin filament) muscle Z-line, beta, CAPZB	285
40	plexin C1, PLXNC1	282
41	nuclear factor I/B, NFIB	279
42	islet amyloid polypeptide, IAPP	276
43	nephroblastoma overexpressed gene, NOV	275
44	hyperpolarization activated cyclic nucleotide-gated potassium channel 4, HCN4	269
45	calsyntenin 2, CLSTN2	268
46	potassium intermediate/small conductance calcium-activated channel, subfamily N, member 1, KCNN1	266
47	sodium channel, voltage-gated, type II, alpha subunit, SCN2A	264
48	neuroligin 1, NLGN1	261
49	ELKS/RAB6-interacting/CAST family member 2, ERC2	261
50	scratch homolog 1, zinc finger protein (Drosophila), SCRT1	252
51	low density lipoprotein receptor-related protein 1, LRP1	249
52	hypothetical protein LOC728392 /// NLR family, pyrin domain containing 1, LOC728392 /// NLRP1	249
53	opiate receptor-like 1, OPRL1	246
54	myosin, heavy chain 14, non-muscle, MYH14	243
55	nitric oxide synthase 1 (neuronal), NOS1	240
56	wingless-type MMTV integration site family, member 2B, WNT2B	238
57	glutamate receptor, metabotropic 1, GRM1	231
58	glutamate receptor interacting protein 1, GRIP1	230
59	myelin associated glycoprotein, MAG	229
60	chemokine (C-C motif) ligand 3 /// chemokine (C-C motif) ligand 3-like 1 /// chemokine (C-C motif) ligand 3-like 3, CCL3 /// CCL3L1 /// CCL3L3	228
61	family with sequence similarity 162, member A, FAM162A	228
62	sphingosine-1-phosphate receptor 5, S1PR5	227
63	protein tyrosine phosphatase, receptor type, R, PTPRR	225
64	IKAROS family zinc finger 1 (Ikaros), IKZF1	225
65	potassium intermediate/small conductance calcium-activated channel, subfamily N, member 3, KCNN3	221
66	solute carrier family 18 (vesicular monoamine), member 2, SLC18A2	219
67	glutamate receptor, ionotropic, *N*-methyl d-aspartate 1, GRIN1	216
68	v-src sarcoma (Schmidt-Ruppin A-2) viral oncogene homolog (avian), SRC	216
69	jagged 1, JAG1	215
70	adenylate cyclase activating polypeptide 1 (pituitary), ADCYAP1	215
71	ATPase, Ca++ transporting, plasma membrane 2, ATP2B2	214
72	tripartite motif-containing 2, TRIM2	213
73	netrin 1, NTN1	212
74	paired related homeobox 1, PRRX1	209
75	purinergic receptor P2X, ligand-gated ion channel, 3, P2RX3	207
76	inhibitor of DNA binding 4, dominant negative helix-loop-helix protein, ID4	203
77	solute carrier family 5 (choline transporter), member 7, SLC5A7	202
78	empty spiracles homeobox 1, EMX1	202
79	muscle, skeletal, receptor tyrosine kinase, MUSK	200
80	GATA binding protein 2, GATA2	193
81	cadherin 13, H-cadherin (heart), CDH13	192
82	Rho/Rac guanine nucleotide exchange factor (GEF) 2, ARHGEF2	191
83	anaplastic lymphoma receptor tyrosine kinase, ALK	191
84	cholecystokinin A receptor, CCKAR	190
85	GLI family zinc finger 2, GLI2	183
86	cholinergic receptor, nicotinic, beta 1 (muscle), CHRNB1	182
87	NK2 homeobox 2, NKX2-2	181
88	purinergic receptor P2X, ligand-gated ion channel, 4, P2RX4	180
89	gamma-aminobutyric acid (GABA) receptor, rho 2, GABRR2	179
90	PDZ and LIM domain 5, PDLIM5	177
91	plasminogen activator, urokinase, PLAU	172
92	cannabinoid receptor 1 (brain), CNR1	172
93	chondrolectin, CHODL	172
94	neurexin 2, NRXN2	171
95	parkinson protein 2, E3 ubiquitin protein ligase (parkin), PARK2	169
96	calcium channel, voltage-dependent, L type, alpha 1F subunit, CACNA1F	168
97	neuregulin 1, NRG1	164
98	zinc finger protein 536, ZNF536	162
99	endothelin 3, EDN3	161
100	paired box 7, PAX7	161
101	calcium/calmodulin-dependent protein kinase II beta, CAMK2B	161
102	solute carrier family 30 (zinc transporter), member 3, SLC30A3	160
103	ciliary neurotrophic factor /// zinc finger protein 91 homolog (mouse) /// ZFP91-CNTF readthrough transcript, CNTF /// ZFP91 /// ZFP91-CNTF	159
104	calcium channel, voltage-dependent, T type, alpha 1I subunit, CACNA1I	156
105	membrane associated guanylate kinase, WW and PDZ domain containing 2, MAGI2	155
106	sigma non-opioid intracellular receptor 1, SIGMAR1	155
107	leptin, LEP	152
108	microtubule-associated protein tau, MAPT	150
109	erythropoietin receptor, EPOR	147
110	frizzled homolog 8 (Drosophila), FZD8	147
111	nuclear mitotic apparatus protein 1, NUMA1	147
112	ninjurin 2, NINJ2	144
113	probable transcription factor PML-like /// promyelocytic leukemia, LOC652346 /// PML	144
114	fasciculation and elongation protein zeta 1 (zygin I), FEZ1	143
115	ribonucleotide reductase M1, RRM1	142
116	retinoic acid receptor, beta, RARB	142
117	metallothionein 3, MT3	142
118	vascular endothelial growth factor A, VEGFA	141
119	glycoprotein M6A, GPM6A	140
120	runt-related transcription factor 1, RUNX1	136
121	cholinergic receptor, nicotinic, delta, CHRND	135
122	testis specific, 10, TSGA10	135
123	growth hormone secretagogue receptor, GHSR	135
124	guanine nucleotide binding protein (G protein), beta polypeptide 3, GNB3	134
125	glycine receptor, beta, GLRB	132
126	runt-related transcription factor 1; translocated to, 1 (cyclin D-related), RUNX1T1	131
127	synaptotagmin V, SYT5	131
128	bridging integrator 1, BIN1	130
129	general transcription factor IIi, GTF2I	128
130	mitogen-activated protein kinase kinase 7, MAP2K7	127
131	peroxisome proliferator-activated receptor gamma, coactivator 1 alpha, PPARGC1A	126
132	v-erb-a erythroblastic leukemia viral oncogene homolog 4 (avian), ERBB4	125
133	retinoic acid receptor, alpha, RARA	123
134	baculoviral IAP repeat-containing protein 1-like /// NLR family, apoptosis inhibitory protein, LOC100510692 /// NAIP	123
135	myosin VA (heavy chain 12, myoxin), MYO5A	122
136	heat shock protein 90kDa alpha (cytosolic), class B member 1, HSP90AB1	121
137	voltage-dependent anion channel 1, VDAC1	120
138	prostaglandin-endoperoxide synthase 2 (prostaglandin G/H synthase and cyclooxygenase), PTGS2	120
139	spectrin, beta, non-erythrocytic 1, SPTBN1	120
140	tubulin, beta 2A /// tubulin, beta 2B, TUBB2A /// TUBB2B	119
141	misshapen-like kinase 1, MINK1	119
142	neural cell adhesion molecule 1, NCAM1	119
143	kelch-like 1 (Drosophila), KLHL1	119
144	sperm associated antigen 9, SPAG9	118
145	gonadotropin-releasing hormone 1 (luteinizing-releasing hormone), GNRH1	116
146	cholinergic receptor, nicotinic, beta 3, CHRNB3	115
147	neuralized homolog (Drosophila), NEURL	115
148	SRY (sex determining region Y)-box 14, SOX14	115
149	purinergic receptor P2X, ligand-gated ion channel, 1, P2RX1	112
150	transcription factor 4, TCF4	112
151	lysozyme, LYZ	111
152	MYC associated factor X, MAX	111
153	synaptojanin 1, SYNJ1	108
154	ret proto-oncogene, RET	108
155	cadherin 2, type 1, N-cadherin (neuronal), CDH2	108
156	AXL receptor tyrosine kinase, AXL	108
157	ataxia telangiectasia mutated, ATM	107
158	parvalbumin, PVALB	107
159	glyceraldehyde-3-phosphate dehydrogenase, GAPDH	107
160	Rap guanine nucleotide exchange factor (GEF) 1, RAPGEF1	106
161	protein kinase C, gamma, PRKCG	106
162	neurofibromin 2 (merlin), NF2	105
163	serrate RNA effector molecule homolog (Arabidopsis), SRRT	105
164	syntaxin 3, STX3	105
165	X-box binding protein 1, XBP1	104
166	potassium large conductance calcium-activated channel, subfamily M, beta member 2, KCNMB2	104
167	chemokine (C-X3-C motif) receptor 1, CX3CR1	104
168	aldehyde dehydrogenase 1 family, member A2, ALDH1A2	103
169	drebrin 1, DBN1	103
170	UDP glycosyltransferase 8, UGT8	103
171	achaete-scute complex homolog 1 (Drosophila), ASCL1	103
172	POU class 4 homeobox 3, POU4F3	102
173	neurofibromin 1, NF1	102
174	steroidogenic acute regulatory protein, STAR	101
175	histamine receptor H3, HRH3	101
176	nuclear receptor subfamily 2, group F, member 6, NR2F6	100
177	transforming growth factor, beta 1, TGFB1	100
178	homeobox D3, HOXD3	100
**DOWN**	**Gene Title**	**Percent Change**
81	5-hydroxytryptamine (serotonin) receptor 3A, HTR3A	−100
82	neuroligin 3, NLGN3	−101
83	aquaporin 1 (Colton blood group), AQP1	−101
84	SH3 and multiple ankyrin repeat domains 2, SHANK2	−102
85	neurochondrin, NCDN	−102
86	astrotactin 1, ASTN1	−102
87	mitogen-activated protein kinase 8 interacting protein 2, MAPK8IP2	−103
88	limbic system-associated membrane protein, LSAMP	−103
89	calcium binding protein 1, CABP1	−106
90	integrin, beta 1 (fibronectin receptor, beta polypeptide, antigen CD29 includes MDF2, MSK12), ITGB1	−107
91	discs, large (Drosophila) homolog-associated protein 2, DLGAP2	−108
92	doublecortin, DCX	−108
93	colony stimulating factor 3 (granulocyte), CSF3	−108
94	advanced glycosylation end product-specific receptor, AGER	−108
95	corticotropin releasing hormone receptor 1, CRHR1	−109
96	neuropeptides B/W receptor 2, NPBWR2	−109
97	even-skipped homeobox 1, EVX1	−110
98	retinoid X receptor, gamma, RXRG	−110
99	cytoplasmic polyadenylation element binding protein 3, CPEB3	−112
100	alpha tubulin acetyltransferase 1, ATAT1	−113
101	paralemmin, PALM	−115
102	tumor necrosis factor, TNF	−115
103	fatty acid binding protein 7, brain, FABP7	−118
104	olfactory marker protein, OMP	−118
105	Amphiregulin, AREG	−118
106	opioid receptor, kappa 1, OPRK1	−119
107	calbindin 2, CALB2	−119
108	phosphodiesterase 10A, PDE10A	−121
109	early growth response 1, EGR1	−121
110	cell cycle exit and neuronal differentiation 1, CEND1	−123
111	5-hydroxytryptamine (serotonin) receptor 3B, HTR3B	−123
112	synaptosomal-associated protein, 23kDa, SNAP23	−123
113	sodium channel, voltage-gated, type XI, alpha subunit, SCN11A	−124
114	growth arrest-specific 7, GAS7	−124
115	contactin 1, CNTN1	−125
116	neuroligin 4, X-linked, NLGN4X	−128
117	gamma-aminobutyric acid (GABA) A receptor, alpha 1, GABRA1	−130
118	leucine zipper, putative tumor suppressor 1, LZTS1	−130
119	mesenchyme homeobox 2, MEOX2	−131
120	TYRO3 protein tyrosine kinase, TYRO3	−131
121	synaptophysin, SYP	−132
122	coiled-coil domain containing 64, CCDC64	−132
123	leucine-rich, glioma inactivated 1, LGI1	−132
124	nerve growth factor receptor, NGFR	−132
125	cholinergic receptor, nicotinic, beta 4, CHRNB4	−135
126	5-hydroxytryptamine (serotonin) receptor 2A, HTR2A	−135
127	myocyte enhancer factor 2C, MEF2C	−138
128	cholinergic receptor, nicotinic, alpha 4, CHRNA4	−139
129	prodynorphin, PDYN	−142
130	discs, large homolog 2 (Drosophila), DLG2	−142
131	neurexin 1, NRXN1	−144
132	secretin, SCT	−148
133	serpin peptidase inhibitor, clade F (alpha-2 antiplasmin, pigment epithelium derived factor), member 1, SERPINF1	−148
134	tachykinin receptor 3, TACR3	−150
135	Ras homolog enriched in brain, RHEB	−150
136	PARK2 co-regulated, PACRG	−153
137	glutamate receptor, ionotropic, kainate 5, GRIK5	−159
138	bone morphogenetic protein 2, BMP2	−159
139	choline O-acetyltransferase, CHAT	−160
140	sodium channel, voltage-gated, type I, alpha subunit, SCN1A	−162
141	TOX high mobility group box family member 3, TOX3	−163
142	gastric inhibitory polypeptide, GIP	−164
143	corticotropin releasing hormone receptor 2, CRHR2	−165
144	kinesin family member 1A, KIF1A	−165
145	RAB35, member RAS oncogene family, RAB35	−166
146	protein kinase C, theta, PRKCQ	−167
147	cell adhesion molecule with homology to L1CAM (close homolog of L1), CHL1	−171
148	unc-51-like kinase 4 (C. elegans), ULK4	−172
149	wingless-type MMTV integration site family, member 4, WNT4	−175
150	thyroid stimulating hormone receptor, TSHR	−175
151	potassium voltage-gated channel, Shal-related subfamily, member 3, KCND3	−175
152	contactin 2 (axonal), CNTN2	−180
153	glutamate receptor, ionotropic, N-methyl D-aspartate 2A, GRIN2A	−180
154	fibronectin leucine rich transmembrane protein 1, FLRT1	−183
155	gamma-aminobutyric acid (GABA) A receptor, gamma 3, GABRG3	−186
156	calcium/calmodulin-dependent protein kinase IG, CAMK1G	−187
157	interleukin 6 receptor, IL6R	−190
158	calsyntenin 3, CLSTN3	−191
159	vesicle-associated membrane protein 1 (synaptobrevin 1), VAMP1	−193
160	promyelocytic leukemia, PML	−196
161	ATPase, H+ transporting, lysosomal accessory protein 2, ATP6AP2	−209
162	mitogen-activated protein kinase 8 interacting protein 3, MAPK8IP3	−209
163	estrogen receptor 2 (ER beta), ESR2	−216
164	cytochrome b-245, beta polypeptide, CYBB	−217
165	purinergic receptor P2Y, G-protein coupled, 11 /// PPAN-P2RY11 readthrough, P2RY11 /// PPAN-P2RY11	−219
166	sonic hedgehog, SHH	−220
167	growth differentiation factor 11, GDF11	−221
168	protein tyrosine phosphatase, receptor type, D, PTPRD	−221
169	ELK1, member of ETS oncogene family, ELK1	−224
170	regulating synaptic membrane exocytosis 1, RIMS1	−225
171	hairy/enhancer-of-split related with YRPW motif-like, HEYL	−228
172	neurotrophic tyrosine kinase, receptor, type 3, NTRK3	−230
173	potassium voltage-gated channel, Shab-related subfamily, member 2, KCNB2	−233
174	regulator of G-protein signaling 6, RGS6	−235
175	glycine receptor, alpha 3, GLRA3	−235
176	potassium voltage-gated channel, shaker-related subfamily, beta member 1, KCNAB1	−235
177	guanine nucleotide binding protein (G protein), alpha transducing activity polypeptide 1, GNAT1	−242
178	proprotein convertase subtilisin/kexin type 2, PCSK2	−242
179	nerve growth factor (beta polypeptide), NGF	−243
180	corticotropin releasing hormone, CRH	−243
181	laminin, alpha 1, LAMA1	−245
182	cyclic nucleotide gated channel alpha 3, CNGA3	−249
183	glutamate receptor, ionotropic, kainate 1, GRIK1	−254
184	lin-28 homolog A (C. elegans), LIN28A	−259
185	empty spiracles homeobox 2, EMX2	−260
186	cyclin-dependent kinase 5, regulatory subunit 1 (p35), CDK5R1	−260
187	agrin, AGRN	−264
188	T-box, brain, 1, TBR1	−272
189	stathmin-like 2, STMN2	−274
190	microcephalin 1, MCPH1	−275
191	ELAV (embryonic lethal, abnormal vision, Drosophila)-like 4 (Hu antigen D), ELAVL4	−282
192	mitogen-activated protein kinase 8 interacting protein 1, MAPK8IP1	−289
193	calcium channel, voltage-dependent, N type, alpha 1B subunit, CACNA1B	−290
194	FEZ family zinc finger 2, FEZF2	−295
195	dopamine receptor D4, DRD4	−296
196	zinc finger E-box binding homeobox 1, ZEB1	−300
197	T-cell leukemia homeobox 1, TLX1	−311
198	sterile alpha motif domain containing 4A, SAMD4A	−315
199	opioid binding protein/cell adhesion molecule-like, OPCML	−333
200	fibroblast growth factor receptor 2, FGFR2	−337
201	SRY (sex determining region Y)-box 1, SOX1	−337
202	neurogenin 1, NEUROG1	−345
203	PTK2B protein tyrosine kinase 2 beta, PTK2B	−348
204	somatostatin receptor 5, SSTR5	−353
205	myelin basic protein, MBP	−361
206	EPH receptor A7, EPHA7	−365
207	G protein-coupled receptor 173, GPR173	−373
208	S100 calcium binding protein A5, S100A5	−374
209	acyl-CoA synthetase long-chain family member 6, ACSL6	−384
210	family with sequence similarity 107, member A, FAM107A	−407
211	Kv channel interacting protein 1, KCNIP1	−413
212	Fas apoptotic inhibitory molecule 2, FAIM2	−416
213	bradykinin receptor B1, BDKRB1	−426
214	discs, large homolog 4 (Drosophila), DLG4	−452
215	adenylate cyclase 10 (soluble), ADCY10	−460
216	cyclin-dependent kinase 5, regulatory subunit 2 (p39), CDK5R2	−481
217	EPH receptor A3, EPHA3	−485
218	phosphodiesterase 1A, calmodulin-dependent, PDE1A	−485
219	chemokine (C-X-C motif) receptor 4, CXCR4	−496
220	membrane metallo-endopeptidase, MME	−540
221	paired-like homeodomain 3, PITX3	−541
222	notch 3, NOTCH3	−547
223	discs, large (Drosophila) homolog-associated protein 1, DLGAP1	−547
224	slit homolog 1 (Drosophila), SLIT1	−553
225	bassoon (presynaptic cytomatrix protein), BSN	−563
226	cadherin, EGF LAG seven-pass G-type receptor 1 (flamingo homolog, Drosophila), CELSR1	−647
227	calcium channel, voltage-dependent, beta 4 subunit, CACNB4	−672
228	necdin homolog (mouse), NDN	−729
229	endothelin receptor type B, EDNRB	−768
230	cholinergic receptor, muscarinic 2, CHRM2	−1049

**Table 13 brainsci-07-00020-t013:** Distribution of Genes Affected by GHK and Associated with Motor Neurons.

Percent Change in Gene Expression	Genes UP	Genes DOWN
50%–99%	9	5
100%–199%	2	0
200%–299%	2	1
300%–399%	0	0
400%–499%	0	2
500%+	0	1
Total	13	9

**Table 14 brainsci-07-00020-t014:** GHK and Genes Associate with Motor Neurons.

**UP**	**Gene Title**	**Percent Change**
1	calcium channel, voltage-dependent, P/Q type, alpha 1A subunit, CACNA1A	286
2	plexin C1, PLXNC1	282
3	GLI family zinc finger 2, GLI2	183
4	NK2 homeobox 2, NKX2-2	181
**DOWN**	**Gene Title**	**Percent Change**
1	slit homolog 1 (Drosophila), SLIT1	−553
2	chemokine (C-X-C motif) receptor 4, CXCR4	−496
3	EPH receptor A3, EPHA3	−485
4	sonic hedgehog, SHH	−220

**Table 15 brainsci-07-00020-t015:** Distribution of Genes Affected by GHK and Associated with Glial Cells.

Percent Change in Gene Expression	Genes UP	Genes DOWN
50%–99%	11	4
100%–199%	7	3
200%–299%	4	4
300%–399%	2	1
400%–499%	0	1
500%+	0	2
Total	24	15

**Table 16 brainsci-07-00020-t016:** GHK and Genes Associated with Glial Cells.

**UP**	**Gene Title**	**Percent Change**
1	neurogenic differentiation 4, NEUROD4	362
2	growth associated protein 43, GAP43	305
3	nuclear factor I/B, NFIB	279
4	caspase 1, apoptosis-related cysteine peptidase (interleukin 1, beta, convertase), CASP1	257
5	Kruppel-like factor 15, KLF15	238
6	adenylate cyclase activating polypeptide 1 (pituitary), ADCYAP1	215
7	neuregulin 1, NRG1	164
8	versican, VCAN	134
9	protein kinase C, eta, PRKCH	124
10	SWI/SNF related, matrix associated, actin dependent regulator of chromatin, subfamily a, member 4, SMARCA4	107
11	chemokine (C-X3-C motif) receptor 1, CX3CR1	104
12	achaete-scute complex homolog 1 (Drosophila), ASCL1	103
13	neurofibromin 1, NF1	102
**DOWN**	**Gene Title**	**Percent Change**
1	necdin homolog (mouse), NDN	−729
2	insulin-like growth factor 1 (somatomedin C), IGF1	−522
3	forkhead box D4 /// forkhead box D4-like 1, FOXD4 /// FOXD4L1	−498
4	PTK2B protein tyrosine kinase 2 beta, PTK2B	−348
5	pleiomorphic adenoma gene 1, PLAG1	−276
6	lin-28 homolog A (C. elegans), LIN28A	−259
7	sonic hedgehog, SHH	−220
8	forkhead box E1 (thyroid transcription factor 2), FOXE1	−204
9	allograft inflammatory factor 1, AIF1	−144
10	GDNF family receptor alpha 2, GFRA2	−141
11	chondroitin sulfate proteoglycan 4, CSPG4	−113

**Table 17 brainsci-07-00020-t017:** Distribution of Gene Affected by GHK and Associated with Astrocytes.

Percent Change in Gene Expression	Genes UP	Genes DOWN
50%–99%	8	3
100%–199%	5	2
200%–299%	2	1
300%–399%	0	0
400%–499%	0	0
500%+	0	0
Total	15	6

**Table 18 brainsci-07-00020-t018:** GHK and Genes Associated with Astrocytes.

**UP**	**Gene Title**	**Percent Change**
1	chemokine (C-C motif) ligand 3 /// chemokine (C-C motif) ligand 3-like 1 /// chemokine (C-C motif) ligand 3-like 3, CCL3 /// CCL3L1 /// CCL3L3	228
2	inhibitor of DNA binding 4, dominant negative helix-loop-helix protein, ID4	203
3	NK2 homeobox 2, NKX2-2	181
4	metallothionein 3, MT3	142
5	bridging integrator 1, BIN1	130
6	matrix metallopeptidase 14 (membrane-inserted), MMP14	114
7	neurofibromin 1, NF1	102
**DOWN**	**Gene Title**	**Percent Change**
1	neurotrophic tyrosine kinase, receptor, type 3, NTRK3	−230
2	contactin 2 (axonal), CNTN2	−180
3	bone morphogenetic protein 2, BMP2	−159

**Table 19 brainsci-07-00020-t019:** Distribution of Genes Affected by GHK and Associated with Schwann Cells.

Percent Change in Gene Expression	Genes UP	Genes DOWN
50%–99%	5	1
100%–199%	2	0
200%–299%	0	0
300%–399%	1	1
400%–499%	0	0
500%+	0	0
Total	8	2

**Table 20 brainsci-07-00020-t020:** GHK and Genes Associated with Schwann Cells.

**UP**	**Gene Title**	**Percent Change**
1	Mediator complex subunit 12, MED12	393
2	neurofibromin 2 (merlin), NF2	105
3	neurofibromin 1, NF1	102
**DOWN**	**Gene Title**	**Percent Change**
1	cytochrome P450, family 11, subfamily A, polypeptide 1, CYP11A1	−393

**Table 21 brainsci-07-00020-t021:** Distribution of Genes Affected by GHK and Associated with Myelin.

Percent Change in Gene Expression	Genes UP	Genes DOWN
50%–99%	24	5
100%–199%	8	8
200%–299%	4	0
300%–399%	0	3
400%–499%	0	2
500%+	0	0
Total	36	18

**Table 22 brainsci-07-00020-t022:** GHK and Genes Associated with Myelin.

**UP**	**Gene Title**	**Percent Change**
1	inositol 1,4,5-triphosphate receptor, type 3, ITPR3	298
2	sodium channel, voltage-gated, type II, alpha subunit, SCN2A	264
3	myelin associated glycoprotein, MAG	229
4	inhibitor of DNA binding 4, dominant negative helix-loop-helix protein, ID4	203
5	aspartoacylase, ASPA	195
6	probable transcription factor PML-like /// promyelocytic leukemia, LOC652346 /// PML	144
7	retinoic acid receptor, beta, RARB	142
8	retinoic acid receptor, alpha, RARA	123
9	myosin VA (heavy chain 12, myoxin), MYO5A	122
10	neurofibromin 1, NF1	102
11	histamine receptor H3, HRH3	101
12	transforming growth factor, beta 1, TGFB1	100
**DOWN**	**Gene Title**	**Percent Change**
1	chemokine (C-X-C motif) receptor 4, CXCR4	−496
2	gap junction protein, gamma 2, 47kDa, GJC2	−428
3	lethal giant larvae homolog 1 (Drosophila), LLGL1	−393
4	myelin basic protein, MBP	−361
5	chromosome 11 open reading frame 9, C11orf9	−342
6	promyelocytic leukemia, PML	−196
7	myelin protein zero, MPZ	−180
8	contactin 2 (axonal), CNTN2	−180
9	toll-like receptor 2, TLR2	−169
10	laminin, alpha 2, LAMA2	−150
11	retinoid X receptor, gamma, RXRG	−110
12	integrin, beta 1 (fibronectin receptor, beta polypeptide, antigen CD29 includes MDF2, MSK12), ITGB1	−107
13	thyroglobulin, TG	−100

**Table 23 brainsci-07-00020-t023:** Distribution of Genes Affected by GHK and Associated with Dendrites.

Percent Change in Gene Expression	Genes UP	Genes DOWN
50%–99%	47	14
100%–199%	19	31
200%–299%	11	15
300%–399%	8	3
400%–499%	0	3
500%+	2	2
Total	87	68

**Table 24 brainsci-07-00020-t024:** GHK and Genes Associated with Dendrites.

**UP**	**Gene Title**	**Percent Change**
1	potassium voltage-gated channel, Shal-related subfamily, member 1, KCND1	845
2	contactin associated protein-like 2, CNTNAP2	581
3	leukocyte specific transcript 1, LST1	395
4	gamma-aminobutyric acid (GABA) A receptor, alpha 5 /// gamma-aminobutyric acid receptor subunit alpha-5-like, GABRA5 /// LOC100509612	392
5	chemokine (C-C motif) ligand 19, CCL19	378
6	doublecortin-like kinase 1, DCLK1	365
7	p21 protein (Cdc42/Rac)-activated kinase 1, PAK1	363
8	potassium voltage-gated channel, Shaw-related subfamily, member 3, KCNC3	332
9	EPH receptor B1, EPHB1	330
10	gamma-aminobutyric acid (GABA) B receptor, 1 /// ubiquitin D, GABBR1 /// UBD	310
11	calcium channel, voltage-dependent, P/Q type, alpha 1A subunit, CACNA1A	286
12	nephroblastoma overexpressed gene, NOV	275
13	obscurin-like 1, OBSL1	263
14	neuroligin 1, NLGN1	261
15	low density lipoprotein receptor-related protein 1, LRP1	249
16	glutamate receptor, ionotropic, kainate 3, GRIK3	246
17	RNA binding protein, fox-1 homolog (*C. elegans*) 2, RBFOX2	245
18	glutamate receptor, metabotropic 1, GRM1	231
19	glutamate receptor interacting protein 1, GRIP1	230
20	glutamate receptor, ionotropic, *N*-methyl d-aspartate 1, GRIN1	216
21	MCF.2 cell line derived transforming sequence, MCF2	202
22	purinergic receptor P2X, ligand-gated ion channel, 4, P2RX4	180
23	synapsin I, SYN1	170
24	Abl-interactor 2, ABI2	168
25	calcium channel, voltage-dependent, L type, alpha 1F subunit, CACNA1F	168
26	membrane associated guanylate kinase, WW and PDZ domain containing 2, MAGI2	155
27	ubiquitin-conjugating enzyme E2I (UBC9 homolog, yeast), UBE2I	150
28	nuclear mitotic apparatus protein 1, NUMA1	147
29	glutamate receptor, ionotropic, *N*-methyl d-aspartate 2C, GRIN2C	146
30	probable transcription factor PML-like /// promyelocytic leukemia, LOC652346 /// PML	144
31	fasciculation and elongation protein zeta 1 (zygin I), FEZ1	143
32	glutamate receptor, metabotropic 7, GRM7	140
33	acetylcholinesterase, ACHE	131
34	retinoic acid receptor, alpha, RARA	123
35	misshapen-like kinase 1, MINK1	119
36	kelch-like 1 (Drosophila), KLHL1	119
37	neuralized homolog (Drosophila), NEURL	115
38	protein kinase C, gamma, PRKCG	106
39	drebrin 1, DBN1	103
40	neurofibromin 1, NF1	102
**DOWN**	**Gene Title**	**Percent Change**
1	bassoon (presynaptic cytomatrix protein), BSN	−563
2	membrane metallo-endopeptidase, MME	−540
3	adenylate cyclase 10 (soluble), ADCY10	−460
4	discs, large homolog 4 (Drosophila), DLG4	−452
5	Kv channel interacting protein 1, KCNIP1	−413
6	EPH receptor A7, EPHA7	−365
7	PTK2B protein tyrosine kinase 2 beta, PTK2B	−348
8	sterile alpha motif domain containing 4A, SAMD4A	−315
9	dopamine receptor D4, DRD4	−296
10	FEZ family zinc finger 2, FEZF2	−295
11	calcium channel, voltage-dependent, N type, alpha 1B subunit, CACNA1B	−290
12	mitogen-activated protein kinase 8 interacting protein 1, MAPK8IP1	−289
13	regulator of G-protein signaling 11, RGS11	−266
14	cyclin-dependent kinase 5, regulatory subunit 1 (p35), CDK5R1	−260
15	glutamate receptor, ionotropic, kainate 1, GRIK1	−254
16	thyroid hormone receptor, alpha (erythroblastic leukemia viral (v-erb-a) oncogene homolog, avian), THRA	−253
17	cyclic nucleotide gated channel alpha 3, CNGA3	−249
18	adenylate cyclase 2 (brain), ADCY2	−247
19	proprotein convertase subtilisin/kexin type 2, PCSK2	−242
20	Rho guanine nucleotide exchange factor (GEF) 15, ARHGEF15	−230
21	potassium voltage-gated channel, Shal-related subfamily, member 3, KCND3	−224
22	protein tyrosine phosphatase, receptor type, D, PTPRD	−221
23	cytochrome b-245, beta polypeptide, CYBB	−217
24	GABA(A) receptors associated protein like 3, pseudogene, GABARAPL3	−197
25	neutrophil cytosolic factor 1C pseudogene, NCF1C	−196
26	promyelocytic leukemia, PML	−196
27	C-reactive protein, pentraxin-related, CRP	−182
28	glutamate receptor, ionotropic, *N*-methyl d-aspartate 2A, GRIN2A	−180
29	tubby like protein 1, TULP1	−176
30	Mitogen-activated protein kinase 8 interacting protein 3, MAPK8IP3	−174
31	cell adhesion molecule with homology to L1CAM (close homolog of L1), CHL1	−171
32	choline *O*-acetyltransferase, CHAT	−160
33	glutamate receptor, ionotropic, kainate 5, GRIK5	−159
34	glutamate receptor, ionotropic, kainate 4, GRIK4	−155
35	5-hydroxytryptamine (serotonin) receptor 6, HTR6	−150
36	tachykinin receptor 3, TACR3	−150
37	5-hydroxytryptamine (serotonin) receptor 5A, HTR5A	−149
38	protease, serine, 12 (neurotrypsin, motopsin), PRSS12	−141
39	cholinergic receptor, nicotinic, alpha 4, CHRNA4	−139
40	5-hydroxytryptamine (serotonin) receptor 2A, HTR2A	−135
41	leucine zipper, putative tumor suppressor 1, LZTS1	−130
42	neuroligin 4, X-linked, NLGN4X	−128
43	glutamate receptor, ionotrophic, AMPA 3, GRIA3	−126
44	glutamate receptor, metabotropic 6, GRM6	−120
45	paralemmin, PALM	−115
46	copine VI (neuronal), CPNE6	−114
47	cytoplasmic polyadenylation element binding protein 3, CPEB3	−112
48	corticotropin releasing hormone receptor 1, CRHR1	−109
49	doublecortin, DCX	−108
50	regulator of G-protein signaling 14, RGS14	−108
51	apolipoprotein E, APOE	−107
52	calcium binding protein 1, CABP1	−106
53	mitogen-activated protein kinase 8 interacting protein 2, MAPK8IP2	−103
54	neurochondrin, NCDN	−102

**Table 25 brainsci-07-00020-t025:** Distribution of Genes Affected by GHK and Associated with Oligodendrocytes.

Percent Change in Gene Expression	Genes UP	Genes DOWN
50%–99%	6	4
100%–199%	6	3
200%–299%	3	1
300%–399%	0	1
400%–499%	0	1
500%+	1	0
Total	16	10

**Table 26 brainsci-07-00020-t026:** GHK and Genes Associated with Oligodendrocytes.

**UP**	**Gene Title**	**Percent Change**
1	tumor protein p73, TP73	938
2	adenylate cyclase activating polypeptide 1 (pituitary), ADCYAP1	215
3	gelsolin, GSN	214
4	inhibitor of DNA binding 4, dominant negative helix-loop-helix protein, ID4	203
5	aspartoacylase, ASPA	195
6	NK2 homeobox 2, NKX2-2	181
7	dopamine receptor D3, DRD3	164
8	histone deacetylase 11, HDAC11	105
9	achaete-scute complex homolog 1 (Drosophila), ASCL1	103
10	neurofibromin 1, NF1	102
**DOWN**	**Gene Title**	**Percent Change**
1	chemokine (C-X-C motif) receptor 4, CXCR4	−496
2	chromosome 11 open reading frame 9, C11orf9	−342
3	sonic hedgehog, SHH	−220
4	zinc finger protein 287, ZNF287	−143
5	early growth response 1, EGR1	−121
6	apolipoprotein E, APOE	−107

**Table 27 brainsci-07-00020-t027:** Distribution of Genes Affected by GHK and Associated with Sensory Nerve Cells.

Percent Change in Gene Expression	Genes UP	Genes DOWN
50%–99%	45	25
100%–199%	24	36
200%–299%	18	6
300%–399%	7	1
400%–499%	1	3
500%+	2	4
Total	97	75

**Table 28 brainsci-07-00020-t028:** GHK and Gene Associate with Sensory Nerve Cells.

**UP**	**Gene Title**	**Percent Change**
1	opioid receptor, mu 1, OPRM1	1294
2	T-box 1, TBX1	553
3	adrenergic, beta-1-, receptor, ADRB1	477
4	gamma-aminobutyric acid (GABA) A receptor, alpha 5 /// gamma-aminobutyric acid receptor subunit alpha-5-like, GABRA5 /// LOC100509612	392
5	calcium channel, voltage-dependent, L type, alpha 1D subunit, CACNA1D	372
6	olfactory receptor, family 2, subfamily W, member 1, OR2W1	370
7	guanine nucleotide binding protein (G protein), alpha activating activity polypeptide, olfactory type, GNAL	366
8	olfactory receptor, family 2, subfamily B, member 6, OR2B6	345
9	cyclic nucleotide gated channel beta 1, CNGB1	330
10	EPH receptor B1, EPHB1	330
11	inositol 1,4,5-triphosphate receptor, type 3, ITPR3	298
12	olfactory receptor, family 7, subfamily A, member 17, OR7A17	285
13	nuclear factor I/B, NFIB	279
14	islet amyloid polypeptide, IAPP	276
15	opiate receptor-like 1, OPRL1	246
16	potassium voltage-gated channel, KQT-like subfamily, member 4, KCNQ4	245
17	myosin, heavy chain 14, non-muscle, MYH14	243
18	taste receptor, type 2, member 13, TAS2R13	237
19	olfactory receptor, family 2, subfamily F, member 2, OR2F2	232
20	glutamate receptor, metabotropic 1, GRM1	231
21	chemokine (C-C motif) ligand 3 /// chemokine (C-C motif) ligand 3-like 1 /// chemokine (C-C motif) ligand 3-like 3, CCL3 /// CCL3L1 /// CCL3L3	228
22	polycystic kidney disease 2-like 1, PKD2L1	225
23	glutamate receptor, ionotropic, *N*-methyl d-aspartate 1, GRIN1	216
24	adenylate cyclase activating polypeptide 1 (pituitary), ADCYAP1	215
25	ATPase, Ca++ transporting, plasma membrane 2, ATP2B2	214
26	olfactory receptor, family 7, subfamily C, member 1, OR7C1	207
27	purinergic receptor P2X, ligand-gated ion channel, 3, P2RX3	207
28	neuropeptide Y receptor Y1, NPY1R	201
29	family with sequence similarity 38, member B, FAM38B	193
30	olfactory receptor, family 1, subfamily A, member 1, OR1A1	189
31	taste receptor, type 2, member 14, TAS2R14	181
32	purinergic receptor P2X, ligand-gated ion channel, 4, P2RX4	180
33	receptor accessory protein 2, REEP2	174
34	endothelin receptor type A, EDNRA	173
35	cannabinoid receptor 1 (brain), CNR1	172
36	melanocortin 1 receptor (alpha melanocyte stimulating hormone receptor), MC1R	164
37	olfactory receptor, family 12, subfamily D, member 3 /// olfactory receptor, family 5, subfamily V, member 1, OR12D3 /// OR5V1	163
38	odorant binding protein 2A /// odorant binding protein 2B, OBP2A /// OBP2B	162
39	prepronociceptin, PNOC	150
40	phospholipase C, beta 2, PLCB2	148
41	glutamate receptor, metabotropic 7, GRM7	140
42	oxytocin, prepropeptide, OXT	136
43	WD repeat domain 1, WDR1	127
44	olfactory receptor, family 1, subfamily D, member 4 (gene/pseudogene) /// olfactory receptor, family 1, subfamily D, member 5, OR1D4 /// OR1D5	125
45	UDP glucuronosyltransferase 2 family, polypeptide A1 /// UDP glucuronosyltransferase 2 family, polypeptide A2, UGT2A1 /// UGT2A2	121
46	prostaglandin-endoperoxide synthase 2 (prostaglandin G/H synthase and cyclooxygenase), PTGS2	120
47	taste receptor, type 2, member 4, TAS2R4	118
48	lysozyme, LYZ	111
49	protein kinase C, gamma, PRKCG	106
50	collagen, type XI, alpha 1, COL11A1	103
51	POU class 4 homeobox 3, POU4F3	102
52	nuclear receptor subfamily 2, group F, member 6, NR2F6	100
**DOWN**	**Gene Title**	**Percent Change**
1	taste receptor, type 2, member 9, TAS2R9	−1494
2	endothelin receptor type B, EDNRB	−768
3	necdin homolog (mouse), NDN	−729
4	membrane metallo-endopeptidase, MME	−540
5	EPH receptor A3, EPHA3	−485
6	arachidonate lipoxygenase 3, ALOXE3	−461
7	bradykinin receptor B1, BDKRB1	−426
8	gap junction protein, beta 4, 30.3kDa, GJB4	−317
9	nerve growth factor (beta polypeptide), NGF	−243
10	guanine nucleotide binding protein (G protein), alpha transducing activity polypeptide 1, GNAT1	−242
11	olfactory receptor, family 3, subfamily A, member 1, OR3A1	−234
12	apelin receptor, APLNR	−230
13	olfactory receptor, family 2, subfamily F, member 1 /// olfactory receptor, family 2, subfamily F, member 2, OR2F1 /// OR2F2	−212
14	olfactory receptor, family 12, subfamily D, member 3, OR12D3	−201
15	olfactory receptor, family 6, subfamily A, member 2, OR6A2	−199
16	cholecystokinin B receptor, CCKBR	−198
17	carbonic anhydrase VI, CA6	−192
18	olfactory receptor, family 5, subfamily I, member 1, OR5I1	−191
19	collagen, type XI, alpha 2, COL11A2	−186
20	olfactory receptor, family 10, subfamily H, member 3, OR10H3	−182
21	glutamate receptor, ionotropic, N-methyl D-aspartate 2A, GRIN2A	−180
22	protein phosphatase, EF-hand calcium binding domain 2, PPEF2	−178
23	sodium channel, nonvoltage-gated 1 alpha, SCNN1A	−175
24	trace amine associated receptor 5, TAAR5	−168
25	gastric inhibitory polypeptide, GIP	−164
26	olfactory receptor, family 2, subfamily H, member 1, OR2H1	−156
27	olfactory receptor, family 2, subfamily J, member 2, OR2J2	−155
28	otoferlin, OTOF	−155
29	discs, large homolog 2 (Drosophila), DLG2	−142
30	cholinergic receptor, nicotinic, alpha 4, CHRNA4	−139
31	5-hydroxytryptamine (serotonin) receptor 2A, HTR2A	−135
32	tectorin alpha, TECTA	−126
33	sodium channel, voltage-gated, type XI, alpha subunit, SCN11A	−124
34	olfactory receptor, family 7, subfamily C, member 2, OR7C2	−120
35	taste receptor, type 2, member 16, TAS2R16	−120
36	glutamate receptor, metabotropic 6, GRM6	−120
37	opioid receptor, kappa 1, OPRK1	−119
38	ATPase, H+ transporting, lysosomal 56/58kDa, V1 subunit B1, ATP6V1B1	−118
39	olfactory marker protein, OMP	−118
40	contactin 5, CNTN5	−116
41	cysteinyl leukotriene receptor 2, CYSLTR2	−113
42	olfactory receptor, family 2, subfamily H, member 2, OR2H2	−110
43	rhodopsin, RHO	−108
44	interleukin 10, IL10	−107
45	olfactory receptor, family 11, subfamily A, member 1, OR11A1	−107
46	polymeric immunoglobulin receptor, PIGR	−107
47	guanine nucleotide binding protein (G protein), gamma 13, GNG13	−106
48	tubby homolog (mouse), TUB	−101
49	glutamate receptor, metabotropic 8, GRM8	−101
50	cystatin S, CST4	−101

**Table 29 brainsci-07-00020-t029:** Distribution of Genes Affected by GHK and Associated with Spinal Nerve Cells.

Percent Change in Gene Expression	Genes UP	Genes DOWN
50%–99%	8	6
100%–199%	9	3
200%–299%	1	2
300%–399%	0	1
400%–499%	1	0
500%+	1	1
Total	20	13

**Table 30 brainsci-07-00020-t030:** GHK and Genes Associated with Spinal Nerve Cells.

**UP**	**Gene Title**	**Percent Change**
1	tumor protein p73, TP73	938
2	smoothened homolog (Drosophila), SMO	415
3	calcium channel, voltage-dependent, P/Q type, alpha 1A subunit, CACNA1A	286
4	GATA binding protein 2, GATA2	193
5	GLI family zinc finger 2, GLI2	183
6	NK2 homeobox 2, NKX2-2	181
7	dopamine receptor D3, DRD3	164
8	paired box 7, PAX7	161
9	slit homolog 3 (Drosophila), SLIT3	154
10	polycystic kidney disease 1 (autosomal dominant), PKD1	137
11	achaete-scute complex homolog 1 (Drosophila), ASCL1	103
12	neurofibromin 1, NF1	102
**DOWN**	**Gene Title**	**Percent Change**
1	slit homolog 1 (Drosophila), SLIT1	−553
2	SRY (sex determining region Y)-box 1, SOX1	−337
3	growth differentiation factor 11, GDF11	−221
4	sonic hedgehog, SHH	−220
5	glutamate receptor, ionotropic, *N*-methyl d-aspartate 2A, GRIN2A	−180
6	even-skipped homeobox 1, EVX1	−110
7	aquaporin 1 (Colton blood group), AQP1	−101

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
