# Peer review of "The Effect of the Human Peptide GHK on Gene Expression Relevant to Nervous System Function and Cognitive Decline"

_brainsci, 2017, doi:10.3390/brainsci7020020_

Round 1

Reviewer 1 Report

Summary:

GHK is a human copper –binding peptide with biological actions that appear to counter age-associated diseases and conditions. GHK which declines with age has beneficial effects on the health of chondrocytes liver cells, human fibroblasts , improves wound healing and tissue regeneration , increases collagen, decorin, angiogenesis and nerve regrowth. GHK was reported as the most active of 1309 bioactive substances capable of reversing the expression of 54 genes in metastatic prone signature. Copper deficiency may contribute to ALS, Huntington’s, Alzheimer’s etc. Virtually all biological GHK effects require presence of copper 2+ chelated to the tripeptide. GHK collagen films raised concentration of copper by 9 fold. GHK may show Regenerative and protective actions, anti-anxiety and anti-pain function and antioxidant activity. Recent studies include synthesis of GHK Cu analogues with higher Anti-ROS activity. Using three gene expression profiles, two from PC3 cells and one from MCF7 cells the investigators examined the genes increased or decreased in expression by GHK.

Major points

1

Could the investigators please indicate if the altered gene expression patterns were the result of a defined concentration or range of concentrations of GHK? How critical are the GHK concentrations likely to be?

2

In the data in tables 9 and 10 and also table 14 how would the investigators propose to understand which responses were primarily related to GHK stimulation and which were more indirect or secondary responses?

3

The investigators should give more detail regarding how the lists of gene expression alterations provided can be further analyzed for future use –either in anticipated clinical applications or in mechanism based studies. Please also indicate if further bioinformatics software and known bioinformatics approaches may be specifically useful in some of these analyses.

Minor points:

In Figure 2 from Reference 41 is the figure reproduced with permission or is it adapted from the previous publication?

Author Response

P { margin-bottom: 0.08in; }

Dear Reviewer,

Thank you for taking time to review the manuscript and give us valuable suggestions. We did our best to revise the manuscript accordingly.

As you advised, we specified GHK’s concentrations used in the Broad Institute experiments. This is a very good point and very important.

In the data in tables 9 and 10 and also table 14 we cannot establish which responses were primarily related to GHK stimulation and which were more indirect or secondary responses. The responses were observed after addition of GHK to cell cultures and compared to control. The addition of GHK resulted in up- or down-regulation of certain genes. GHK was used without copper, however, since GHK has high affinity for copper and there is always some copper in cell growth media, the effects could be related to GHK-Cu. Further studies are needed to identify the exact mechanism of the observed effects.

You asked to give more detail regarding how the lists of gene expression alterations provided can be further analyzed in the future. We clarified our position in the conclusion. We believe there should be parallel studies on further exploring gene effects using modern bioinformatics software as well as conducting more laboratory and clinical studies on GHK’s effects on nervous system function. Most GHK research has been focused on wound healing and skin/hair health and its effects on the nervous system function only recently started attracting attention. Considering a large number of genes which GHK affects and whose expression can affect nervous system functioning, this has to change. There is a great potential for GHK as a treatment for neurodegenerative disorders and it has to be explored.

There was a question about Figure 2 from Reference 41 - is the figure reproduced with permission or is it adapted from the previous publication? The figure is from Lymphokines 1983,8, 425–446 article and remains Dr. Pickart’s property. We made a note about it.

We also checked English spelling and grammar.

We hope that with the inclusion of the proposed revisions the manuscript is ready to be published.

Sincerely,

Dr. Pickart and co-authors

Reviewer 2 Report

brainsci-167488

The Effect of the Human Peptide GHK on Gene Expression Relevant to Nervous System Function and Cognitive Decline

Loren Pickart, Jessica Michelle Vasquez-Soltero, Anna Margolina

This very comprehensive review of the biology of the peptide GHK (Gly-His-Lys) represents an enormous effort by the authors. For a reader well educated in the field of peptide biology, this is an understandable but somewhat overwhelming amount of information, which needs to be filtered to become useful. One of the problems is the enormous tables which (as presented for review) are each many pages long and make reading the actual text nearly impossible (e.g. mitochondrial genes, 5 pages; neuronal genes, 12 pages). Another problem is the inclusion of offhand comments more appropriate to everyday speech and unsourced, such as (lines 247-8) the comment about Mixed Martial Arts fighters injecting GHK before fights.  This kind of comment runs the risk of creating a public health problem, since it can be taken out of context on social media and become a new “fad”.

One way to make the Tables more useful to the reader is to streamline them. One suggestion is to eliminate some as not adding much to the story, such as (line 254) Table 2, since Table 3 has all the actual information in Table 2. Care must be exercised in the long Tables. A quick perusal in (line 257) Table 3 found CCKAR listed as Cholecystokinin, when in fact it is the CCK-A Receptor. Most of the Gene Names in Table 3 are indeed gene names, so it would be most consistent to list GRIA3 as the gene name and list AMPA3 in the comments, and for consistency list CB1 in the comments for CNR1. This reviewer did not rake through the very long tables, but suspects similar problems exist in those tables.

Author Response

P { margin-bottom: 0.08in; }

Dear Reviewer,

Thank you for taking time to review the manuscript and give us valuable feedback. We did our best to revise the manuscript accordingly.

We agree that the manuscript contained long tables with overwhelming amount of data. We reduced the amount of data by 1) including only those genes whose expression change was +/-100% or more and by removing all mitochondria data (5 pages), which can be included in a separate paper. We also double-checked the description of the genes and their abbreviations for accuracy.

We removed mentioning of Cage fighters and other anecdotal information. We agree it may be a safety concern.

We clarified our position on what can be done with this quite extensive data we collected. We believe there should be parallel studies on further exploring gene effects as well as conducting more laboratory and clinical studies on GHK’s effects on nervous system function. Most GHK research has been focused on wound healing and skin/hair health and its effects on the nervous system function only recently started attracting attention. Considering a large number of genes which GHK affects and whose expression can affect nervous system functioning, this has to change. There is a great potential for GHK as a treatment for neurodegenerative disorders and it has to be explored.

We hope that with the inclusion of the proposed revisions the manuscript is ready to be published.

Sincerely,

Dr. Pickart and co-authors

Round 2

Reviewer 1 Report

The major points are appropriately addressed.

Clarification should be given on whether the authors own figure 2 from reference 41 or if the figure belongs to the previous publisher. If the publisher owns it then the figure should be revised to be a modified version of the original (adapted) or else should be presented with permission from the publisher.

Author Response

Reviewer 1 indicated that the major points are appropriately addressed. 

Reviewer 2 Report

fine as resubmitted

Author Response

The "figure 2" is from my research (Pickart). However, we did have the figure redrawn and modified as slightly different so that the would be permission issues.